# Host Feeding Patterns of *Mansonia* (Diptera, Culicidae) in Rural Settlements near Porto Velho, State of Rondonia, Brazil

**DOI:** 10.3390/biom13030553

**Published:** 2023-03-17

**Authors:** Diego Peres Alonso, Jandui Almeida Amorim, Tatiane Marques Porangaba de Oliveira, Ivy Luizi Rodrigues de Sá, Fábio Sossai Possebon, Dario Pires de Carvalho, Kaio Augusto Nabas Ribeiro, Paulo Eduardo Martins Ribolla, Maria Anice Mureb Sallum

**Affiliations:** 1Biotechnology Institute and Bioscience Institute, Sao Paulo State University, Botucatu 18618-689, Brazil; 2Departamento de Epidemiologia, Faculdade de Saúde Pública, Universidade de São Paulo, São Paulo 01246-904, Brazil; 3Santo Antônio Energia, Porto Velho 76805-812, Brazil

**Keywords:** *Mansonia* spp., blood source, NGS, biting activity

## Abstract

Mosquito females of the genus *Mansonia* (Blanchard) can be a nuisance to humans and animals since they are voraciously hematophagous and feed on the blood of a variety of vertebrates. Despite their relevance, there is a lack of investigation into the blood-feeding patterns of the *Mansonia* species. Knowledge of the host preference is crucial in establishing the public health importance of a mosquito species and its potential to be involved in the transmission dynamics of pathogens. Species that are primarily anthropophilic can be more effective in spreading vector-borne pathogens to humans. In this study, we used an Illumina Nextera sequencing protocol and the QIIME2 workflow to assess the diversity of DNA sequences extracted in the ingested blood of mosquito species to evaluate the overall and local host choices for three species: *Ma. titillans*, *Ma.* Amazonensis, and *Ma. humeralis*, in rural areas alongside the Madeira River in the vicinities of the Santo Antonio Energia (SAE) reservoir in the municipality of Porto Velho, Rondônia, Western Brazil. By performing our analysis pipeline, we have found that host diversity per collection site showed a significant heterogeneity across the sample sites. In addition, in rural areas, *Ma. amazonensis* present a high affinity for *B. taurus*, *Ma. humeralis* shows an overall preference for *C. familiaris* and *B. taurus*, but also *H. sapiens* and *E. caballus* in urban areas, and *Ma. titillans* showed more opportunistic behavior in rural areas, feeding on wild animals and *G. gallus*, though with an overall preference for *H. sapiens*.

## 1. Introduction

Mosquitoes of the genus *Mansonia* (Blanchard) are classified into the *Mansonioides* (Theobald) subgenus which encompasses species found in Asia and Africa [1], and the *Mansonia* subgenus with species that occur in the Neotropical Region, extending to central and southern Nearctic Region [2,3]. *Mansonia* females are voraciously hematophagous, feeding on the blood of a wide variety of vertebrates. When present in high population densities, these mosquitoes can cause disturbances to humans and domestic animals, deaths of small livestock, and stress in large livestock. Blood feeding is essential for mosquito ovarian development, egg development, and the spread of pathogens among hosts. The blood feeding pattern is evaluated by the blood meal analysis, and it can be influenced by environmental and biological factors of the mosquito species population [4]. Linked to their blood feeding, mosquito females are associated with the spread of pathogens, such as arboviruses, nematodes, and *Plasmodium* protozoa [5], and they can shape the transmission cycle of pathogens across a community of vertebrate hosts. Species of the *Mansonia* genus such as *Mansonia indubitans* and *Mansonia titillans* are effective vectors of the Venezuelan equine encephalitis virus (VEEV) in Peru [6] and Venezuela [6,7]. In addition, some *Mansonia* species were found naturally infected with the Mayaro virus [8].

*Mansonia* species expand locally in areas of permanent and semipermanent water collections with floating macrophytes and aquatic vegetation. Host plants for the immature stages of *Mansonia* include *Eichhornia crassipes*, *Limnobium laevigatum*, *Pistia stratiotes*, and *Salvinia molesta* [9]. The association between aquatic macrophytes and *Mansonia* is a biological and essential feature for the reproductive cycle of species. Egg batches, larval stages, and pupa attach themselves to submerged plant roots using their siphon adapted to perforate and get oxygen from auriferous aerenchyma [3]. An abundance of aquatic plants is related to a decrease in the water flow and increased presence of organic residues that can cause water eutrophication that contributes to the increase and dispersion of these mosquitos in the environment [2].

Species of the genus *Mansonia* feed preferentially at night, peaking at dawn and dusk [10]. The females readily bite and blood-feed on humans and domestic and wild vertebrate hosts [11,12], usually those that are close to the habitats of the immatures [13]. In addition, artificial light attracts females that feed on blood in outdoor environments [14]. Several parameters can influence the presence, density, dispersion, and blood-feeding behavior of *Mansonia*, and other mosquitos [4]. Blood feeding is influenced by a complex system of interrelated biological and environmental factors, such as blood host availability, environmental conditions, deforestation [15], changes in the use of land, and the water ecosystem distribution for agricultural expansion [16,17]. Mosquito species that are generalist-feeders [11,12] are resilient to both human-dominated environments and changes in blood host communities can be favored, becoming abundant and dominant.

The host preference is determined genetically [4]. It shows that mosquito species can feed on a particular species or group, independent of the host abundance in the environment [5]. Despite variations in feeding behavior and host preference, the knowledge of blood-feeding patterns can help to establish which species are involved in the transmission dynamics of a mosquito-borne pathogen [18,19]. Mosquitoes have evolved different host preferences due to selection and adaptations to the environment, interactions with other organisms, limitations imposed by evolutionary history, and adaptive foraging behavior [20]. Although many mosquito species are generalists with a hierarchical preference for hosts [21,22], others exhibit a strong preference for specific hosts [22].

The mosquito species that feed preferentially on humans can be an effective vector in spreading human pathogens. *Anopheles gambiae* s.s., the primary vector of human *Plasmodium* across the sub-Saharan region in Africa, and *Aedes aegypti*, the primary vector of dengue, are primarily anthropophilic [23,24]. As a result, these species are more likely to blood feed on humans than other hosts, improving the likelihood of the pathogen’s survival and transmission [25]. Interestingly, both species, *An. gambiae* and *Ae. aegypti*, have evolved from generalists to human-feeding specialists [26,27].

Both intrinsic and extrinsic factors affect mosquito host preference. Intrinsic factors are determined by the physiology of the mosquitoes [21] and can be driven by selection and therefore have a genetic background [28]. Despite a genetic basis, high plasticity mediated by the density of host species characterizes blood-feeding preference [29]. The abundance of hosts is a readily accessible source of blood for mosquitoes.

As mentioned previously, females of the genus *Mansonia* can be a nuisance to humans and animals since they are voraciously hematophagous and feed on the blood of a variety of vertebrates. Despite their relevance, there is a lack of investigation focusing on the blood-feeding patterns of species of the *Mansonia* subgenus. Since the host preference is a key factor in establishing the public health importance of a mosquito species and its potential to be involved in the transmission of pathogens, we proposed in this study a next-generation sequencing (NGS) protocol to assess the diversity of DNA sequences extracted in the ingested blood of *Mansonia* species. NGS has allowed both deep sequencing and shotgun-style metagenomics to be used more widely. To date, few studies have applied NGS technologies to the identification of bloodmeals taken by field-collected arthropods. The results showed that NGS is sensitive enough to amplify traces of DNA from mosquito midguts and can be employed to identify the vertebrate species in multiple host blood meals [30,31].

In this study, we used the Illumina Nextera sequencing protocol and the QIIME2 workflow to (one) assess the diversity of DNA sequences extracted from the blood ingested by field-collected females of *Mansonia* species and (two) evaluate the overall and local host choices of Ma. titillans, Ma. Amazonensis, and Ma. humeralis, in rural areas alongside the Madeira River in the vicinities of the Santo Antonio Energia (SAE) reservoir in the municipality of Porto Velho, Rondônia, Western Brazil.

## 2. Materials and Methods

### 2.1. Study Areas and Field Collections

Madeira is the second largest Andean river in the Amazon basin and the main tributary of the Amazon River. Originally, the Madeira River region in the municipality of Porto Velho, Rondônia state was a seasonally flooded environment, with predominant vegetation composed of lowland and submontane open rainforest, freshwater swamps along the river basin, small tributaries rivers, terra firme forest, igapó (flooded) forest, and riverine communities. Recently, the floodplain forest vegetation along the Madeira River was cleared for the construction of two hydroelectric run-of-the-river dams in the municipality of Porto Velho. The new dam system caused permanent flooding along a large stretch upstream of the dam in the Madeira River [32], eliminating part of the floodplain forest [33], and leading to a 47.2% increase in the flooded area between the two dams [32].

The study areas comprised major habitats such as the Madeira River and its tributaries, freshwater swamps along the river basin, and disturbed tropical rainforest fragments with varied forest cover percentages intermixed with small family subsistence farms, large livestock farms, such as cattle, swine, and chicken, agricultural land, and rural communities. This region has a warm humid equatorial climate, with a clear delimitation between the rainy and dry seasons. According to the Köppen classification, the climate is AW—rainy tropical with an average temperature varying from 21 to 34 °C and average rainfall varying from 17 to 264 mm monthly. The rainy season is from October to April and the dry season is from June to August, with transition periods in May and September [34].

To identify the blood-feeding behavior of the *Mansonia* species, engorged females were collected outdoors in distinct rural locations in the vicinities of Porto Velho, Rondônia, Brazil. The sampled areas are along a 70 km section of the Madeira River from the Jaci-Parana district to approximately 20 km west of Porto Velho town (Figure 1). Engorged females were collected from 5 sampling sites in the floodplain area on the right margin of the Madeira River (1 (*n* = 46), 2 (*n* = 82), 3 (*n* = 107), 4 (*n* = 90), and 5 (*n* = 118)) (Appendix A) (Figure 1).

Female collections were carried out using the barrier screen sampling (BSS) method following the protocol employed by [35] from 18h00 until 6h00. Collections were undertaken in March 2020 at the end of the rainy season, when the river water levels were highest. The barrier screen was constructed from a grey mesh fiberglass window screen, 2 m high and 12 m long, and was placed outdoors within ~ 20 m of houses, between houses, and potential oviposition/resting sites to intercept mosquitoes. Mosquitoes were collected using a manual power aspirator by two collectors that were protected by the clothing and hats that protected them from mosquito bites. Collections were conducted for 10 min on both sides of the screen, and collectors moved away for 20 min. The BSS was examined for females at 20 min intervals. Mosquitoes were aspirated from both sides of the screen and euthanized with ethyl acetate (C_4_H_8_O_2_) vapors twice per hour. Samples were stored immediately in plastic containers with silica gel, separated by date, location, peridomestic habitat, and hour of collection. Prior to DNA extraction, all specimens were morphologically identified to the species level using Forattini’s [2] identification key. Females were visually separated as blood-fed or unfed and those fed females were bisected in two, head plus thorax/abdomen, using a sterile entomological pin. Specimens were transferred to individual plastic vials and stored at −80 °C until genomic DNA extraction.

### 2.2. Sample Preparation, PCR Conditions for Ingested Blood Meal Identification, and Sequencing

Genomic DNA extraction was performed individually from the abdomen of the engorged female using a DNeasy Blood & Tissue Kit (Qiagen, Hilden, Germany) according to the manufacturer’s recommendations. The PCR amplification of a 130-base pair (bp) fragment of the mitochondrial 16S gene was carried out using panmammalian and bird primers [36] that were further changed to include Illumina adapter sequences in the 5′region of PCR primers (Illumina16Smam1-F 5′ TCGTCGGCAGCGTCAGATGTGTATAAGAGACAGCGGTTGGGGTGACCTCGGA and Illumina16Smam1-R 5′ GTCTCGTGGGCTCGGAGATGTGTATAAGAGACAGGCTGTTATCCCTAGGGTAACT). Each PCR was performed in a total volume of 25 µL, containing 0.2 µM of each primer, 12.5 µL of GoTaq Master Mix (Promega, Madison, WI, USA), 1 µL of DNA template, and the ultrapure water to reach the final volume. Cycling conditions were: 5 min at 95 °C, 40 cycles of 12 s at 95 °C, 30 s at 59 °C, 25 s at 70 °C, and 7 min at 70 °C. PCR products were visualized on an agarose gel and cleaned with AMPure XP beads. A second round of PCR was performed for Illumina indexing, following the Illumina 16S Metagenomic Sequencing Library Preparation protocol, using the Nextera XT index kit.

DNA quantification was performed by fluorometric quantitation using a Thermo Fisher Scientific QuBit dsDNA HS Assay Kit, according to the manufacturer’s recommendations. All 443 DNA samples were then arranged in three different pools containing 169, 169, and 105 samples respectively. Each pool was loaded on a MiSeq Nano flow cell and sequenced using the MiSeq (Illumina, San Diego, CA, USA) in a 151-cycle paired-end run. Sequence-quality analysis was performed using the FastQC program, and reads were used if results from all analysis modules were approved without errors.

### 2.3. Ingested Blood Analysis

Ingested blood diversity analysis was performed using QIIME2 version 2021.11. Briefly, low-quality sequences and chimeras were discarded using the DADA2 pipeline with standard parameters. Alfa rarefaction and beta diversity were performed with a sampling of 1000 reads from each sample. The representative sequences obtained as the output from the DADA2 pipeline were taxonomically identified through comparative sequencing analysis using blastn algorithm with only the sequences with mitochondrial blast hits being kept to Bray–Curtis dissimilarity abundance analysis. Permutational multivariate analysis of variance (PERMANOVA) between geographical groups and *Mansonia* species was performed with QIIME2 based on Bray–Curtis dissimilarity index distances. After pairwise permutations, values with *p* < 0.05 were considered statistically significant. For each female specimen sequenced, the results were expressed as the total counts of each sequence identified in the ingested blood (Appendix A).

## 3. Results

### 3.1. Sequencing

The Illumina sequencing generated 2,277,583 raw 130 base pairs reads. For each specimen sequenced employing the 16S PCR protocol, we obtained approximately 5000 reads, representing an average of 7.5× coverage.

### 3.2. Ingested Blood Meal Analysis

First, we aimed to assess the overall diversity of the ingested blood for the *Mansonia* species (Figure 2). We identified 17 different host species, regardless of the *Mansonia* species from which the ingested blood was sequenced. The most abundant species identified was *Canis lupus familiaris* (40%), followed by *Bos taurus* (28%), *Homo sapiens* (21%), *Equus caballus* (6%), *Gallus gallus* (1%), *Dasypus* spp., *Pecari tajacu*, *Procyon* sp., *Myoprocta pratti*, *Cervidae*, *Bradypus tridactylus*, *Tamandua mexicana*, *Choloepus* spp., *Hydrochoerus hydrochaeris*, *Sus scrofa*, *Bradypus variegatus*, and *Myrmecophaga tridactyla* (<1%).

Next, the overall diversity of the ingested blood in a broader scenario using the principal coordinates analysis (PCoA) based on the Bray–Curtis index (Figure 3) was verified. To do that, based on the total number of reads obtained, we considered only one host species identified per sample (Appendix A). It is noteworthy to mention that 110 samples out of 443 presented only one blood meal source. From the analysis, it is possible to verify the clustering of samples based on the different sources of ingested blood. There were three major clusters (*B. taurus*, *C. Familiaris,* and *H. sapiens*) shared by samples, reflecting different proportions of ingested blood from these hosts. The other scattered clusters depict minor distinct hosts found in the blood meal analysis for *Mansonia*.

After these overall analyses, we evaluated host diversity per collection site to check homogeneity in the source of ingested blood, regardless of *Mansonia* species at the different sampling points. The PERMANOVA analysis was performed for pairwise comparisons between localities. The results showed that except for site one × site five and site four × site three, all other comparisons presented consistent dissimilarities regarding the species identified for *Mansonia* ingested blood (Table 1). This result corroborates that the distribution of hosts for *Mansonia* and is statistically different across the localities. A greater proportion of *H. sapiens* was found in site five and site one, compared to a high prevalence of *C. familiaris* in site two, while in the other sampling sites, *S. Scrofa, B. Taurus, G. Gallus,* and *E. caballus*, and wild (all other hosts grouped together from now on) animals (Figure 4) were identified.

Results of the analyses focused on the host diversity indicated a significant heterogeneity among collection points. To understand the differences found, we investigated the local abundance of *Mansonia* hosts regarding *Mansonia* species to reach an accurate estimate of the host preference of *Mansonia* across all collection points. We used PERMANOVA analysis to perform pairwise comparisons between *Mansonia* species collected in each locality (Table 2).

We observed that in site five, all comparisons between *Mansonia* species were significantly different regarding ingested blood species composition. In addition, in site three, *Ma. Humeralis* × *Ma. Titillans,* and *Ma. amazonensis* × *Ma. humeralis* comparisons were significantly different, while in site four, only *Ma. amazonensis* × *Ma. humeralis* comparison presented the same pattern. These findings are shown in Figure 5, where the ingested blood for *Ma. humeralis* was found proportionally distributed among *H. sapiens, Bos taurus, C. Familiaris,* and *E. caballus*, compared to the other two species in site five. For site three, *Ma. humeralis* was found in greater proportion, having *C. familiaris* as a blood meal source compared to the other two species. For site four, the same pattern was observed for both *Ma. humeralis*, and *Ma. amazonensis.*

## 4. Discussion

Identification of blood-meal hosts in blood-sucking arthropod vectors is of major importance to the analysis of both vectorial capacity and the effectiveness of vector control measures. Immunological methods such as the precipitin test, latex agglutination, and enzyme-linked immunosorbent assay (ELISA) were the earliest methods used in mosquito blood-meal analysis [37,38]. These methods have produced interesting results though lately they have been substituted by nucleic acid-based methods. In cases where mosquito populations are known to feed on a limited range of host species, conventional multiplex PCR with species-specific primers [39] or quantitative PCR (qPCR) with species-specific probes [40,41] are the techniques of choice. In this study, we used 16S rRNA as the molecular marker of choice mainly due to its ability to recover heavily degraded vertebrate host DNA found in partially digested bloodmeal. Moreover, despite the small fragment size, a large portion of the diversity of 16S comes from indels, making this critical information in the separation of samples that can be harnessed for speciation [42].

Recently, NGS has allowed both deep sequencing and shotgun-style metagenomics to be used more widely. Metabarcoding approaches paired with NGS to determine the species represented in mixed biological samples offers the opportunity to broadly examine the sequence data from individual samples, including the identification of pathogens, hosts, and vector from a single NGS [43]. To date, few studies have applied NGS technologies to the identification of bloodmeals taken by field-collected arthropods. The results showed that NGS is sensitive enough to amplify traces of DNA from mosquito midguts and accurate enough to reveal multiple-host mosquito and kissing bug meals correctly [30,31]. In addition, the results confirmed that NGS can clearly unveil the evidence of multiple blood feeding with the different proportions of sequenced reads, whereas Sanger sequencing can disclose only the dominant blood source without the quantitative data. Taken together, these aspects demonstrate the advantages of NGS in characterizing vector host preference, which in turn, facilitates vector incrimination, giving valuable insights into zoonotic transmission networks and dynamics of vector-borne pathogens, especially in endemic areas.

The results of our proposed NGS analysis pipeline, focused on the identification of the blood-feeding behavior of the *Mansonia* species, broadly, show that the species sampled can feed on a variety of vertebrate hosts, including human, domestic, and wild animals (17 vertebrate host species in total) (Figure 2). In a recent study, *Nyssorhynchus darlingi* females were sampled in 34 peridomestic habitats in 27 rural communities from 11 municipalities in the Brazilian Amazon states of Acre, Amazonas, Pará, and Rondônia and had their ingested blood analyzed by Sanger sequencing [33]. When compared to our *Mansonia* pipeline analysis, DNA sequence comparison detected only nine vertebrate host species, which is nearly half the amount of species herein detected (17 in total). In addition, several samples were excluded from the analysis due to unreadable sequences, which could be indicating mixed blood meals in those females [33]. The results obtained for the *Mansonia* females that were analyzed indicate the generalistic behavior of the species and the greater sensitivity of NGS sequencing compared to conventional Sanger sequencing.

To our knowledge, this is the first study to explore the QIIME2 workflow to assess the diversity of DNA sequences generated from the ingested blood of mosquito species. The PCoA based on the Bray–Curtis index is useful for investigating the clustering pattern of the sources of blood meals of females, even in a scenario of meals involving multiple hosts. Overall, we found that over 75% of the females analyzed (Appendix A) had fed on more than one blood host. Unfortunately, by using this approach, it is not possible to differentiate consecutive blood meals involving different hosts from multiple feedings in different hosts within the same gonotrophic cycle.

After running the pipeline, we found that the host diversity had a significant heterogeneity across the collection sites, with a greater proportion of *H. sapiens* in site one and site five, whilst a high prevalence of *C. familiaris* was identified in site two. In other sampling sites, the hosts were heterogeneous and included both domestic and wild animals (Figure 4). Interestingly, sites one, two, and five are close to urban areas. This proximity can explain the high abundance of *H. sapiens* blood found in the *Mansonia* species. In addition, the high prevalence of dog blood is certainly indirect evidence of humans in those sites.

To carry out the analyses considering both the local abundance of *Mansonia* hosts and *Mansonia* species, we excluded *Ma. indubitans* since only seven females were collected. The results obtained were statistically significant (Figure 5). In rural locations three and four, *Ma. Amazonensis* blood-fed mainly on *B. Taurus* blood, reaching almost 70% of all specimens analyzed. *Mansonia humeralis* fed more often on *C. Familiaris* and *B. Taurus*, besides *H. Sapiens* and *E. Caballus* in urban areas, whilst in *Ma. Titillans,* we identified an opportunistic feeding behavior, including wild animals and *G. gallus*, though with an overall preference for *H. sapiens.*

A previous study was conducted in the same region we sampled and *Ma. titillans* was the most abundant species collected outdoors and indoors, accounting for up to 77% of all specimens collected [10]. Our findings are corroborated by [10] since the results of the analysis focused on verifying the host distribution for each *Mansonia* species within each locality separate, we found a high relative abundance of *Ma. titillans* that fed on human blood. The preference for human blood was also clear when we compared *Ma*. *Titillans* and *Ma. humeralis* (Figure 5). Also, when the mosquito feeding behavior was verified considering the geographical locations, *Ma. titillans* showed the highest proportion of human blood in site three (Figure 5) which is far from highly human-populated areas. This is additional evidence that *Ma. titillans* has a narrow spectrum of blood hosts.

*Mansonia* species have a wide dispersal range and may fly over marshes, ponds, and lakes to find oviposition sites or vertebrate hosts for blood feeding [44]. Despite being capable of dispersing more than 2 km from the emergence sites, the adult dispersion pattern of the *Mansonia* species is primarily defined by random brief flights. Males and females usually rest on vegetation patches 30 to 100 m far from their immature habitats [45]. Also, environment heterogeneities can influence on a vector species behavior, such as host choice, peak biting time, and resting behavior [46,47]. For instance, studies of *Anopheles arabiensis* showed that heterogeneity in host availability significantly affects both female blood feeding and resting behavior [48]. *Nyssorhynchus darlingi* biting behavior in the Brazilian Amazon is associated with indoor environmental conditions, such as temperature, resting places, and host availability. The behavioral plasticity in specimens that share the same genotype is a key factor that shapes *Ny. darlingi* peak biting time [47].

It is widely accepted that in the Brazilian Amazon, rural settlements, such as those linked to agriculture expansion, forest degradation, and increased urbanization are more likely to present a greater abundance of mosquito species that are resilient to new ecological conditions. In addition, the resilient species are usually habitat opportunistic and host generalist. This is especially true for anopheline species involved in the transmission of *Plasmodium* spp. in a scenario of frontier malaria [49]. Also, urbanization is the major driving force of the behavioral shift towards human biting in *Ae. aegypti.* The human-biting behavior of *Ae. aegypti* originally evolved as a byproduct of the species association with man-made recipients that were the habitats available to survive the long and hot dry season [50].

We can hypothesize that *Mansonia* populations in the area studied might thrive in lentic habitats alongside the Madeira River where aquatic macrophyte plants are abundant to stand *Mansonia* species that depend on the macrophytes for their development and survivorship [51]. Although *Mansonia* is found in great abundance in less degraded landscapes [52,53], direct and indirect anthropic changes in the environment may have generated an adequate landscape for oviposition, development of immature stages, and the increased blood host availability that are necessary to support *Mansonia* populations. The new ecological conditions led to an increased abundance of macrophyte plants and the subsequent *Mansonia* dispersion across the landscapes, where human activities were intensified, including the establishment of domestic animals, livestock, and poultry farming.

## 5. Conclusions

This study provided a comprehensive analysis of the host preference of field-collected females of *Mansonia* species in the vicinities of Porto Velho, Rondônia, Western Brazil. This is the first report using an Illumina Nextera sequencing protocol coupled with the QIIME2 workflow to assess the diversity of DNA sequences extracted in the ingested blood of mosquito species to evaluate the overall and local host choices of *Ma. titillans*, *Ma. amazonensis*, and *Ma. humeralis*. By performing our pipeline, we assessed the blood meal analyses and found that host diversity showed a significant heterogeneity across the collection sites. In rural areas, *Ma. amazonensis* presented a high affinity for *B. taurus*, whereas *Ma. humeralis* fed more frequently on *C. familiaris* and *B. taurus*, though on *H. sapiens* and *E. caballus* in urban areas. *Mansonia titillans* is opportunistic in rural areas, feeding on wild animals and *G. gallus*, with an overall preference for *H. sapiens*. Results presented here also showed that anthropogenic changes in the natural landscapes that favored the shift from lotic to lentic environments, and the changes in land use are driving a cascade of ecological changes, favoring macrophyte plants and *Mansonia* mosquitoes that can coexist with them.

## Figures and Tables

**Figure 1 biomolecules-13-00553-f001:**
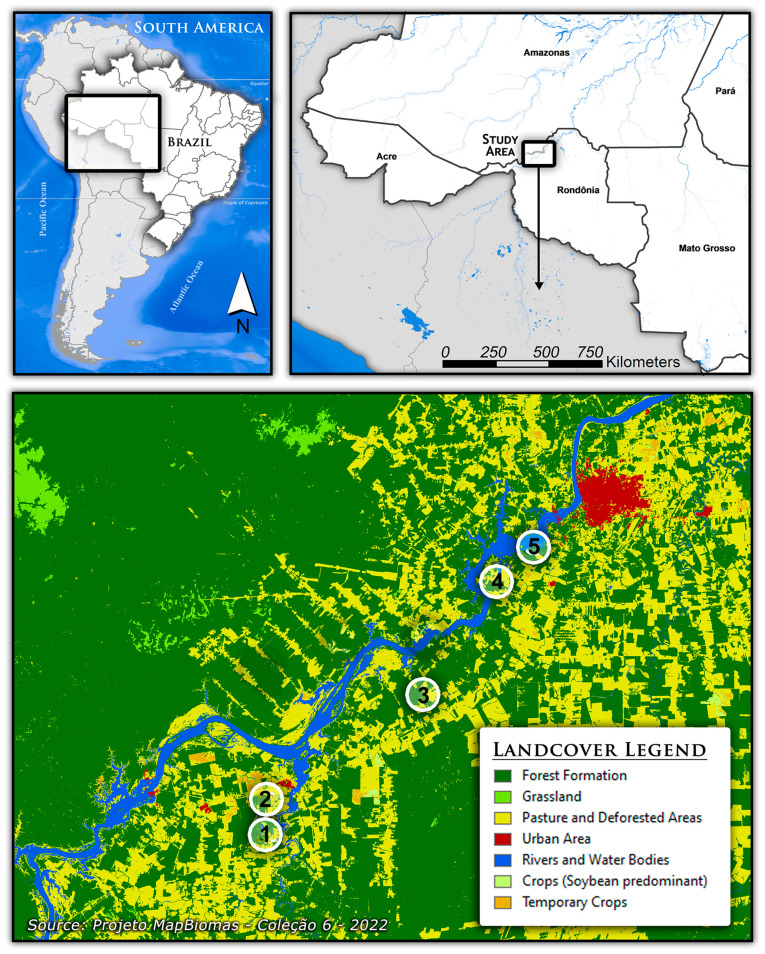
Specimen collections in five localities in the vicinities of the Porto Velho municipality, Rondônia state, Brazil. The numbers represent the collection sites: 1—Jaci Paraná River; 2—Samaúma, Jaci Paraná; 3—Santa Rita settlement; 4—Nova Teotônio village; 5—São Domingos settlement.

**Figure 2 biomolecules-13-00553-f002:**
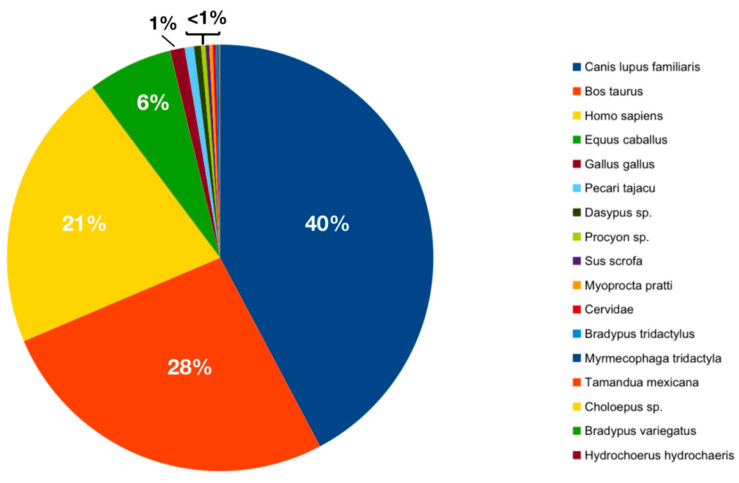
Pie chart representing the overall proportion of ingested blood from different hosts in all *Mansonia* specimens collected.

**Figure 3 biomolecules-13-00553-f003:**
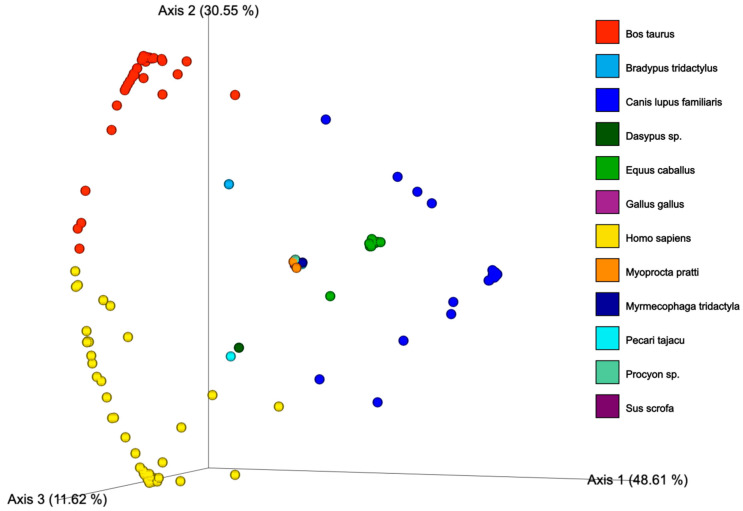
Principal coordinates analysis (PcoA) plot based on Bray–Curtis (quantitative) distance index of *Mansonia* according to the identified blood.

**Figure 4 biomolecules-13-00553-f004:**
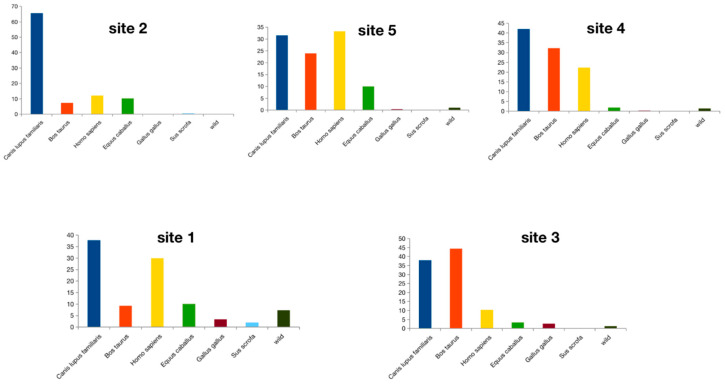
Bar charts representing the proportion of ingested blood from different hosts for *Mansonia* in different collection points. Values in bold denote statiscrical significance at the *q* < 0.05 level.

**Figure 5 biomolecules-13-00553-f005:**
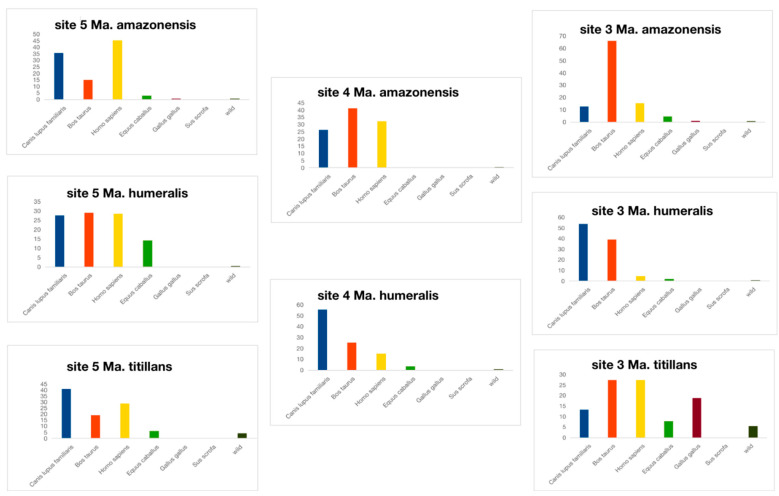
Bar charts representing the proportion of ingested blood from different hosts for each *Mansonia* species sampled in sites 3, 4, and 5. Only the species that were found statistically significant for the comparisons within each locality are represented in the figure (according to Table 2).

**Table 1 biomolecules-13-00553-t001:** Results of PERMANOVA pairwise distances multivariate analysis. Variance of pairwise distance comparisons according to the geographical origin of samples. Values in bold denote statiscrical significance at the *q* < 0.05 level.

Comparisons	Sample Size	Permutations	Pseudo-F	*p*-Value	*q*-Value
site 1 × site 2	126	999	6.978800169	0.001	**0.00250**
site 1 × site 5	162	999	0.90541316	0.468	0.46800
site 2 × site 5	196	999	13.14182445	0.001	**0.00250**
site 4 × site 1	134	999	3.841688365	0.013	**0.01625**
site 4 × site 2	168	999	9.909787075	0.001	**0.00250**
site 4 × site 5	204	999	4.209660355	0.012	**0.01625**
site 4 × site 3	195	999	1.078676499	0.308	0.34222
site 3 × site 1	153	999	7.138065671	0.002	**0.00333**
site 3 × site 2	187	999	15.4967486	0.001	**0.00250**
site 3 × site 5	223	999	9.598897081	0.002	**0.00333**

**Table 2 biomolecules-13-00553-t002:** Results of PERMANOVA pairwise distances multivariate analysis. Variance of pairwise distance comparisons, according to *Mansonia* species at each collection point. Values in bold denote statiscrical significance at the *q* < 0.05 level.

Comparisons per Group	Sample Size	Permutations	Pseudo-F	*p*-Value	*q*-Value
site 2					
*Ma. amazonensis* × *Ma. humeralis*	71	999	1.003637873	0.373	0.4303
*Ma. humeralis* × *Ma. titillans*	69	999	0.5366033	0.599	0.6305
*Ma. amazonensis* × *Ma. titillans*	12	999	1.070379585	0.32	0.3798
site 5					
*Ma. amazonensis* × *Ma. humeralis*	98	999	9.008857428	0.001	**0.0060**
*Ma. amazonensis* × *Ma. titillans*	48	999	4.454286977	0.01	**0.0352**
*Ma. humeralis* × *Ma. titillans*	84	999	10.78372428	0.001	**0.0060**
site 4					
*Ma. amazonensis* × *Ma. humeralis*	78	999	4.128920366	0.02	**0.0489**
*Ma. amazonensis* × *Ma. titillans*	39	999	0.395281343	0.693	0.7107
*Ma. humeralis* × *Ma. titillans*	59	999	1.185069354	0.324	0.3798
site 3					
*Ma. humeralis* × *Ma. titillans*	75	999	6.113456897	0.003	**0.0150**
*Ma. amazonensis* × *Ma. humeralis*	91	999	13.35434213	0.001	**0.0060**
*Ma. amazonensis* × *Ma. titillans*	48	999	2.109241498	0.109	0.1791
site 1					
*Ma. humeralis* × *Ma. titillans*	44	999	1.125252545	0.265	0.33120

## Data Availability

In The datasets generated and analyzed during the current study will be deposited in the Sequence Read Archive (SRA) linked to the PRJNA942296 BioProject identifier.

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
