# Peer review of "Host Feeding Patterns of Mansonia (Diptera, Culicidae) in Rural Settlements near Porto Velho, State of Rondonia, Brazil"

_biomolecules, 2023, doi:10.3390/biom13030553_

Round 1

Reviewer 1 Report (Previous Reviewer 2)

Dear authors,

The manuscript has been significantly improved with more detailed discussions as previously suggested. However, it would be more appropriate if the authors can clarify the nomenclature of each collection site. To prevent ambiguity of localities, the reference codes should be listed with ordinal numbers, and then I would suggest the authors change and rerun these if the samples included from these trapping sites are not redundant with other previous host preference studies.

Since the research using Illumina MiSeq for bloodmeal analysis is still limited, it would be better to explain the rationale and advantages of selecting16S rRNA as a target for host identification, compared to other loci as formerly published.

It would be nice if the authors can comparatively discuss the results with the Sanger sequencing data previously done in Mansonia or other mosquito species in the same investigated areas.

The scientific names in the reference list need to be all italicized. Please carefully check and correct.

All the best

Author Response

Reviewer #1

Dear authors,

The manuscript has been significantly improved with more detailed discussions as previously suggested. However, it would be more appropriate if the authors can clarify the nomenclature of each collection site. To prevent ambiguity of localities, the reference codes should be listed with ordinal numbers, and then I would suggest the authors change and rerun these if the samples included from these trapping sites are not redundant with other previous host preference studies.

Reply: We changed the nomenclature of the collection sites as suggested. We also reinforce that there are no other sites being analyzed or considered to be used in other publications. Field collections of specimens employed for blood feeding analyses were carried out in the sites pointed out in Figure 1.

Since the research using Illumina MiSeq for bloodmeal analysis is still limited, it would be better to explain the rationale and advantages of selecting16S rRNA as a target for host identification, compared to other loci as formerly published.

Reply: We have added a paragraph to explain the advantages of employing Illumina MiSeq to sequence the 16s rRNA gene compared to other methods.

It would be nice if the authors can comparatively discuss the results with the Sanger sequencing data previously done in Mansonia or other mosquito species in the same investigated areas.

Reply: We presented and discussed the results of Sanger sequencing from a study of Nyssorhynchus darlingiin the Brazilian Amazon in comparison to our findings using Illumina MiSeq.

The scientific names in the reference list need to be all italicized. Please carefully check and correct.

Reply: We checked and fixed all the scientific names in the references. Thanks for letting us know.

Reviewer 2 Report (New Reviewer)

The manuscript, Host feeding patterns of Mansonia (Diptera, Culicidae) in rural 2 settlements near Porto Velho, State of Rondonia, Brazil, describes the characterization of host bloodfeeding patterns of Mansonia in a rural area in Brazil by Alonso et al., describes differential blood host profiles in Mansonia species of Brazil analyzed using Next Generation Sequencing and running the data following QIIME2 pipeline.

I think the manuscript reads well and is easy to comprehend except the introduction section. The introduction section needs more revision to show what the goals of the study were or what the hypothesis were. It also needs a paragraph or more to show how Next Generation Sequencing (NGS) is utilized in other studies because the authors are claiming that they are the first to utilize NGS and QIIME2 for mosquito blood meal analysis. I suggest that authors should be careful when they claim that they are the first for a technique in the study field because I have seen some published articles where NGS was utilized in blood meal analysis. The methods section contains excellent description about the study area. It describes well how the study was conducted. The results section presents detailed analysis and the discussion section seems congruent to the results.

Overall, I recommend a minor revision for publication.

Following are some minor suggestions which may improve the manuscript:

(1)   Line 37, change italicized “subgenus” to regular fonts.

(2)   Line 45, insert words, such as, between pathogens and arboviruses

(3)   Line 45, the word, nematodes, appears to be a different font type or a font size from the fonts in other parts of the manuscript.

(4)   Line 52, place a comma (,) between the words, floating macrophytes, and aquatic vegetation.

(5)   Lines 54 – 55, place an “and” between the words, biological and essential.

(6)   Line 72, remove “)”.

(7)   Line 72, Specify what “the species” indicate. It is not clear which species the authors were referring.

(8)   Line 240, typo- “thaIn” should be “that in”

(9)   Line 265, typo, change “have” to ”has”

Author Response

Reviewer #2

The manuscript, Host feeding patterns of Mansonia (Diptera, Culicidae) in rural 2 settlements near Porto Velho, State of Rondonia, Brazil, describes the characterization of host bloodfeeding patterns of Mansonia in a rural area in Brazil by Alonso et al., describes differential blood host profiles in Mansonia species of Brazil analyzed using Next Generation Sequencing and running the data following QIIME2 pipeline.

I think the manuscript reads well and is easy to comprehend except the introduction section. The introduction section needs more revision to show what the goals of the study were or what the hypothesis were. It also needs a paragraph or more to show how Next Generation Sequencing (NGS) is utilized in other studies because the authors are claiming that they are the first to utilize NGS and QIIME2 for mosquito blood meal analysis. I suggest that authors should be careful when they claim that they are the first for a technique in the study field because I have seen some published articles where NGS was utilized in blood meal analysis. The methods section contains excellent description about the study area. It describes well how the study was conducted. The results section presents detailed analysis and the discussion section seems congruent to the results.

Overall, I recommend a minor revision for publication.

Reply: We added a paragraph to explain the rationale of the study hypothesis and presented some studies that used NGS to assess bloodmeal content in mosquitoes. We also clarified that the novelty of our study is to use bloodmeal NGS analysis coupled with QIIME pipeline analysis.

Following are some minor suggestions which may improve the manuscript:

(1)   Line 37, change italicized “subgenus” to regular fonts.

(2)   Line 45, insert words, such as, between pathogens and arboviruses

(3)   Line 45, the word, nematodes, appears to be a different font type or a font size from the fonts in other parts of the manuscript.

(4)   Line 52, place a comma (,) between the words, floating macrophytes, and aquatic vegetation.

(5)   Lines 54 – 55, place an “and” between the words, biological and essential.

(6)   Line 72, remove “)”.

(7)   Line 72, Specify what “the species” indicate. It is not clear which species the authors were referring.

(8)   Line 240, typo- “thaIn” should be “that in”

(9)   Line 265, typo, change “have” to ”has”

Reply: We checked and fixed all the grammar issues. Thanks for pointing our errors.

This manuscript is a resubmission of an earlier submission. The following is a list of the peer review reports and author responses from that submission.

Round 1

Reviewer 1 Report

The resubmitted manuscript has improved. However authors should still look carefully at some points.

1. The nomenclature of collection sites is not clear. Are these numbers random? Does the author omit data from other collection sites? Why weren't they numbered from 1 to 5? If there are more collection points, why were they not included in this study? If there are more sites, they should be reported in this study.

2. Results presented in tables do not need to be repeated in figures. They may be included as supplementary material.

3. The quality of some figures, such as figure 5, is not appropriate. However, if it represents the same as the table, repetition is not necessary. The quality of figure 3 should also be improved. Colors must be standardized between figures.

4. The word study is repeated several times in the conclusion, which makes the text unattractive for the reader.

5. The characteristics of the collection sites could be summarized in a table to facilitate the understanding of the presented discussion. 

Author Response

Reviewer 1

  1. The nomenclature of collection sites is not clear. Are these numbers random? Does the author omit data from other collection sites? Why weren't they numbered from 1 to 5? If there are more collection points, why were they not included in this study? If there are more sites, they should be reported in this study.

Reply:

First of all, thanks for the feedback on this matter. The results reported in this manuscript are part of a broader project on population genetics and metagenomic analysis of Mansonia spp. The nomenclature of the collection sites is the reference code that were used to process the different samples. For this study, we collected mosquitos only in the sites RO7, RO9, RO11, RO12, and RO23, as mentioned in the materials and methods section.

  1. Results presented in tables do not need to be repeated in figures. They may be included as supplementary material.

Reply:

The table presents only the statistics for pairwise comparisons according to Mansonia species at each collection point and the geographical location of samples. We used the figures to show the proportion of different hosts for Mansonia spp. We believe that tables and figures complement each other and need to stay in the main text.

  1. The quality of some figures, such as figure 5, is not appropriate. However, if it represents the same as the table, repetition is unnecessary. The quality of figure 3 should also be improved. Colors must be standardized between figures.

Reply:

Thank you for your suggestion. We changed figure 3 to match the colors of all the other figures. For size constraints we cannot add high resolution figures in the main text of the manuscript, but we provided them separate for final editing.

  1. The word study is repeated several times in the conclusion, which makes the text unattractive for the reader.

Reply:

We apologize for that. The text was corrected as suggested.

  1. The characteristics of the collection sites could be summarized in a table to facilitate the understanding of the presented discussion. 

Thank you for your valuable suggestion. Unfortunately, we do not have detailed information regarding each collection site. Because of lack of accurate data, we decided not to include characteristics of each collection site.

Reviewer 2 Report

Dear authors,

This manuscript has demonstrated the application of the Illumina MiSeq NGS and specialized pipelines for characterizing the blood meal sources of Mansonia mosquitoes. In overall, it was well written and clearly described. However, I have some comments that would be useful for the readers as follows:

1.) Remove the incomplete phrase 'In addition to the presence of blood hosts.' (Line 64)

2.) The platform for detecting the blood meal sources and the rationale should be mentioned in the objective of this project in the introduction (Line 95-98).

3.) As presented in the pie chart (Figure 2), the percentages representing the proportion of blood meal sources are missing.

4.) However, blood meal analysis can be evaluated using immunological methods, such as precipitin, ELISA, as well as molecular methods such as multiplex PCR, Sanger Sequencing. The authors should mention this and discuss more about the superiority of the NGS to the aforementioned methods in blood meal characterization. The NGS can clearly unveil the evidence of multiple blood feeding with the different proportions of sequenced reads whereas Sanger sequencing can reveal only the dominant blood source without the quantitative data.

5.) Additionally, the use of NGS application for characterizing the blood meals has been applied before with several vector species using other genetic markers, for instance, 16s rRNA in Anopheles, Cytochrome b in Triatomine bugs, 12s rRNA in Rhodnius. It would be more informative if the authors would include this references in the discussion of manuscript.

6.) In summary, the authors should emphasize the advantages of NGS helpful for characterizing the host preference of vectors, which greatly facilitate the vector incrimination and fulfill the advanced knowledge of zoonotic transmission networks and transmission dynamics of vector-borne pathogens, especially in the endemic areas.

7.) The information of Illumina index primers for the second round PCR should also be informed. The authors should describe more how to pool the samples into three different pools (169, 169, and 105).

All the best

Author Response

  1. ) Remove the incomplete phrase 'In addition to the presence of blood hosts.' (Line 64)

Reply:

We apologize for our error. The text was corrected as suggested.

  1. ) The platform for detecting the blood meal sources and the rationale should be mentioned in the objective of this project in the introduction (Line 95-98).

Thank you for your comment. We changed the text as suggested.

3.) As presented in the pie chart (Figure 2), the percentages representing the proportion of blood meal sources are missing.

Reply:

We apologize for that. We replaced the figure as you suggested.

4.) However, blood meal analysis can be evaluated using immunological methods, such as precipitin, ELISA, as well as molecular methods such as multiplex PCR, Sanger Sequencing. The authors should mention this and discuss more about the superiority of the NGS to the aforementioned methods in blood meal characterization. The NGS can clearly unveil the evidence of multiple blood feeding with the different proportions of sequenced reads whereas Sanger sequencing can reveal only the dominant blood source without the quantitative data.

Additionally, the use of NGS application for characterizing the blood meals has been applied before with several vector species using other genetic markers, for instance, 16s rRNA in Anopheles, Cytochrome b in Triatomine bugs, 12s rRNA in Rhodnius. It would be more informative if the authors would include this references in the discussion of manuscript.

 In summary, the authors should emphasize the advantages of NGS helpful for characterizing the host preference of vectors, which greatly facilitate the vector incrimination and fulfill the advanced knowledge of zoonotic transmission networks and transmission dynamics of vector-borne pathogens, especially in the endemic areas.

Reply:

Thank you so much for you great feedback. We added two paragraphs in the Discussion section to handle the advantages of the NGS in detecting the host blood meals in the insect vectors, in the light of our proposed analysis pipeline.

5.) The information of Illumina index primers for the second round PCR should also be informed. The authors should describe more how to pool the samples into three different pools (169, 169, and 105).

Reply:

The Illumina index primers are part of the Nextera XT that we employed to prepare the libraries for sequencing. We added this information in the text. Besides that, the pooling process was better described.

Round 2

Reviewer 1 Report

In this second submission, the authors made a significant effort. Overall, the manuscript has been improved. However I still do not recommend publishing this study in its present form.

Dividing a broad project into different publications, as suggested by the authors in one of their answers, is not appropriate and does not justify not reorganizing the current version and correctly identifying the collection sites. The project should be properly mentioned in the manuscript, as well as scientific documents already published and future collection sites that will be explored in future works.